# GROUNDING GRAPH NETWORK SIMULATORS USING PHYSICAL SENSOR OBSERVATIONS

**Jonas Linkerhägner**[1]*       **Niklas Freymuth**[1]       **Paul Maria Scheikl**[1,2]

**Franziska Mathis-Ullrich**[1,2]                              **Gerhard Neumann**[1]

[1]**Institute for Anthropomatics and Robotics,**
Karlsruhe Institute of Technology, Karlsruhe, Germany

[2]**Department Artificial Intelligence in Biomedical Engineering,**
Friedrich-Alexander-University Erlangen-Nürnberg, Erlangen, Germany

## ABSTRACT

Physical simulations that accurately model reality are crucial for many engineering disciplines such as mechanical engineering and robotic motion planning. In recent years, learned Graph Network Simulators produced accurate mesh-based simulations while requiring only a fraction of the computational cost of traditional simulators. Yet, the resulting predictors are confined to learning from data generated by existing mesh-based simulators and thus cannot include real world sensory information such as point cloud data. As these predictors have to simulate complex physical systems from only an initial state, they exhibit a high error accumulation for long-term predictions. In this work, we integrate sensory information to *ground* Graph Network Simulators on real world observations. In particular, we predict the mesh state of deformable objects by utilizing point cloud data. The resulting model allows for accurate predictions over longer time horizons, even under uncertainties in the simulation, such as unknown material properties. Since point clouds are usually not available for every time step, especially in online settings, we employ an imputation-based model. The model can make use of such additional information only when provided, and resorts to a standard Graph Network Simulator, otherwise. We experimentally validate our approach on a suite of prediction tasks for mesh-based interactions between soft and rigid bodies. Our method results in utilization of additional point cloud information to accurately predict stable simulations where existing Graph Network Simulators fail.

## 1 INTRODUCTION

Mesh-based simulation of complex physical systems lies at the heart of many fields in numerical science and engineering (Liu et al., 2022; Reddy, 2019; Rao, 2017; Sabat & Kundu, 2021). Applications include structural mechanics (Zienkiewicz & Taylor, 2005; Stanova et al., 2015), electromagnetics (Jin, 2015; Xiao et al., 2022; Coggon, 1971), fluid dynamics (Chung, 1978; Zawawi et al., 2018; Long et al., 2021) and biomedical engineering (Van Staden et al., 2006; Soro et al., 2018), most of which traditionally depend on highly specialized task-dependent simulators. Recent advancements in deep learning brought rise to more general learned dynamic models such as Graph Network Simulators (GNSs) (Sanchez-Gonzalez et al., 2018; 2020; Pfaff et al., 2021). GNSs learn to predict the dynamics of a system from data by encoding the system state as a graph and then iteratively computing the dynamics for every node in the graph with a Graph Neural Network (GNN) (Scarselli et al., 2009; Battaglia et al., 2018; Wu et al., 2020b). Recent extensions include long-term fluid flow predictions (Han et al., 2022) and dynamics on different scales (Fortunato et al., 2022). Yet, these approaches assume full knowledge of the initial system state, making them ill-suited for applications

---

*correspondence to `jonas.linkerhaegner@alumni.kit.edu`

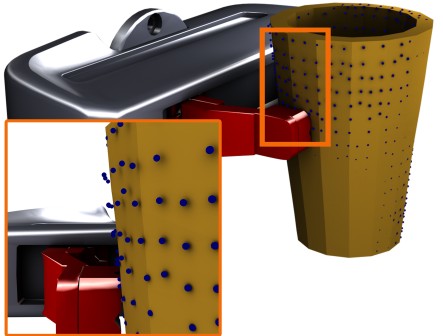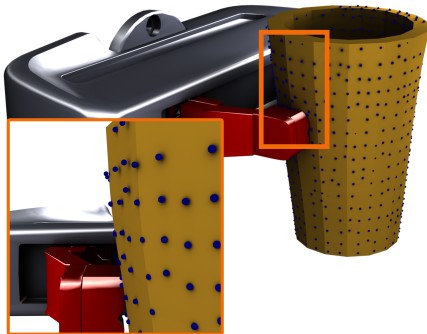

Figure 1: A robot's end-effector (grey, red) grasps a 3-dimensional deformable cavity. The robot maintains an internal simulated prediction of the cavity (orange) for two consecutive simulation steps (left, right). This prediction can deviate from the true state of the cavity over time due to an accumulation of error. However, the true cavity state can infrequently be observed from point cloud data (blue), which the model can use to correct its prediction. Here, the point cloud is used to contract the simulated cavity at the bottom and extend it at the top, causing the points to better align with the mesh surface. We repeat the point cloud from the earlier simulation step in both images for clarity.

like model-predictive control (Camacho & Alba, 2013; Schwenzer et al., 2021) and model-based Reinforcement Learning (Polydoros & Nalpantidis, 2017; Moerland et al., 2020) where accurate predictions must be made based on partial initial states and observations.

In this work, we present Grounding Graph Network Simulators (GGNSs), a new class of GNS that can process sensory information as input to *ground* predictions in the scene observations. More precisely, we extend the graph of the current system state with point cloud data before predicting the system dynamics from it. Since point clouds do not provide correspondences over time, it is difficult to learn dynamics from point clouds alone. Thus, we use mesh-based data to learn the general system dynamics and utilize point clouds to correct the predictions. As the sensory data is not always available, particularly not for future predictions, our architecture is trained with imputed point clouds, i.e., for each time step the model receives point clouds only with a certain probability. This training scheme allows the model to efficiently integrate the additional information whenever provided. During inference, the model iteratively predicts the next system state, using point clouds whenever available to greatly improve the simulation quality, especially for simulations with incomplete initial state information. Furthermore, our architecture addresses a critical research topic for GNSs by alleviating common challenges such as *drift* and error accumulation during long-term predictions.

As a practical example, consider a robot grasping a deformable object. For optimal planning of the grasp, the robot needs to model the state of the deformable object over time and predict the influence of interactions between object and gripper. This prediction not only depends on the initial shape of the object, but also on the forces the robot applies, the kind of material to grasp and external factors such as the temperature, making it difficult to accurately predict how the material will deform over time. However, once the robot starts deforming the object, it may easily observe the deformations in the form of e.g., point clouds. These observations can then be integrated into the state prediction, i.e., they can *ground* the simulation whenever new information becomes available. An example is given in Figure 1. Such observation-aided prediction is similar in nature to e.g., Kalman Filters (Kalman, 1960; Jazwinski, 1970; Becker et al., 2019) as the belief of the system state is updated based on partial observations about the system. However, while Kalman Filters explicitly integrate novel information into the belief in a mathematical fashion, we instead simply provide this information to a learned model as additional unstructured sensor input.

We evaluate GGNS on a suite of 2d and 3d deformation prediction tasks created in the Simulation Open Framework Architecture (SOFA) (Faure et al., 2012). Comparing our approach to an existing GNS (Pfaff et al., 2021), we find that adding sensory information in the form of point clouds to our model improves the simulation quality for all tasks. We investigate this behavior through extensive ablation studies, showing the importance of different parameter choices and design decisions. Code and data can be found under `https://github.com/jlinki/GGNS`.

Our list of contributions is as follows: (I) We extend the GNS framework to include sensory information to *ground* predicted simulations in observations of the system state, allowing for accurate

predictions of the full simulation. (II) We propose a simple but effective imputation training scheme that naturally integrates sensory information to GNSs whenever available without substantially increasing training cost or model complexity. (III) We construct and experiment on different deformation prediction tasks and find that the inclusion of sensory information improves performance in all settings, and that it is particularly crucial when the initial system state is not fully known.

## 2 RELATED WORK

***Learned Physics Simulation.*** In recent years there has been a steady increase in research concerning deep learning for physical simulations. Early work in physical reasoning aims at teaching systems to understand physical relations on N-body systems (Battaglia et al., 2016) and deformable objects (Mrowca et al., 2018). A more direct approach is to instead train a learnable simulator from data provided by some existing ground truth simulator. Here, Convoluational Neural Networks (CNNs) have been extensively studied for fluid flow simulation (Tompson et al., 2017; Chu & Thuerey, 2017; Ummenhofer et al., 2020; Kim et al., 2019; Xie et al., 2018) and aerodynamic flow fields (Guo et al., 2016; Zhang et al., 2018; Bhatnagar et al., 2019). Further approaches use standard neural networks for liquid splash simulations (Um et al., 2018) and latent space physics simulation (Wiewel et al., 2019). Such learned physics simulators are considerably faster than their ground-truth counterparts, and that they are usually fully differentiable. Thus, they have been applied to model-based Reinforcement Learning (Mora et al., 2021) and for Inverse Design problems (Baqué et al., 2018; Durasov et al., 2021; Allen et al., 2022b) .

***Graph Network Simulators.*** Graph Network Simulators (GNS) (Sanchez-Gonzalez et al., 2020) are a special case of learned physics simulators that utilize GNNs (Scarselli et al., 2009) to efficiently encode the graph-like structure of many physical problems. They have found wide-spread application in calculating atomic forces (Hu et al., 2021), particle-based simulations (Li et al., 2019; Sanchez-Gonzalez et al., 2020) and mesh-based simulations (Pfaff et al., 2021; Weng et al., 2021; Han et al., 2022; Fortunato et al., 2022; Allen et al., 2022a). Other works in this field directly solve partial differential equations (Alet et al., 2019), and integrates explicit domain knowledge into the learned simulator to improve the predictions (de Avila Belbute-Peres et al., 2020; Li & Farimani, 2021; 2022). Similarly, CNNs have been used to predict particle masses from images to subsequently simulate physical systems with a GNN (Li et al., 2020) via visual grounding. This approach assumes access to a series of images to predict particles and their behavior, whereas GGNS integrates sensor observations into an existing mesh-based simulation. The work most closely related to our research is MeshGraphNet (MGN) (Pfaff et al., 2021), which combines a graph-based encoding of the system state with the next-step prediction of dynamic quantities to produce realistic predictions of mesh-based simulations.

***Simulation from Observation.*** Another variant of learned physics simulation is simulation from observation. Learning directly from observations instead of a ground truth simulator requires less expert knowledge for the design of the simulator and is more applicable to real-world scenarios. Different approaches exist for this type of simulation, including Physical reasoning (Li et al., 2020) and particle-based simulation (Martinkus et al., 2021). Point clouds have been used in CNN-based simulation (Watters et al., 2017; Wang et al., 2019), and combined with PointNet (Charles et al., 2017; Qi et al., 2017) to predict object deformations purely from observational data (Park et al., 2021). Further approaches make use of GNNs to predict object relations (Fetaya et al., 2018) and future frames in a point cloud sequence (Gomes et al., 2021).

***Simulation of Deformable Objects.*** Simulating deformable objects is crucial for many applications such as robotic manipulation tasks (Sanchez et al., 2018). Yet, recent approaches do not take explicit deformation into account (Matas et al., 2018), or only consider highly simplified geometries such as ropes (Sundaresan et al., 2020) or a square piece of cloth (Wu et al., 2020a; Lin et al., 2020; 2022). One reason for this is the high computational cost of existing simulators, which may be alleviated by fast and accurate learned simulators (Pfaff et al., 2021; Weng et al., 2021). Another recent work trains the parameters of a differentiable simulator to align its simulations with real-world observations of deformable objects based on point cloud information (Sundaresan et al., 2022). In this work, we instead utilize point cloud information to improve upon existing mesh-based GNSs in settings where additional point cloud data is available.

## 3 FOUNDATIONS

### 3.1 MESSAGE PASSING NETWORK

Let $\mathcal{G} = (\mathbf{V}, \mathbf{E}, \mathbf{X_V}, \mathbf{X_E})$ be a directed graph with nodes $\mathbf{V}$, edges $\mathbf{E} \subseteq \mathbf{V} \times \mathbf{V}$, node features $\mathbf{X_V} : \mathbf{V} \to \mathbb{R}^{d_\mathbf{V}}$ of dimension $d_\mathbf{V}$ and edge features $\mathbf{X_E} : \mathbf{E} \to \mathbb{R}^{d_\mathbf{E}}$ of dimension $d_\mathbf{E}$. A Message Passing Network (MPN) (Sanchez-Gonzalez et al., 2020; Pfaff et al., 2021) is a GNN consisting of $L$ *Message Passing Blocks* that receives the graph $\mathcal{G}$ as input and outputs a learned representation for each node $\mathbf{V}$ and edge $\mathbf{E}$. Each block $l$ computes updated features for all nodes $v \in \mathbf{V}$ and edges $e \in \mathbf{E}$ as

$$\mathbf{x}_e^{l+1} = f_\mathbf{E}^l(\mathbf{x}_v^l, \mathbf{x}_u^l, \mathbf{x}_e^l), \text{ with } e = (u, v) \text{ and } \mathbf{x}_v^{l+1} = f_\mathbf{V}^l(\mathbf{x}_v^l, \bigoplus_{\{e=(v,u)\in\mathbf{E}\}} \mathbf{x}_e^{l+1}),$$

where $\mathbf{x}_v^0$ and $\mathbf{x}_e^0$ are embeddings of the initial node and edge features of $\mathcal{G}$ and $\oplus$ is a permutation-invariant aggregation such as a sum, max, or mean operator. Furthermore, each $f_.^l$ is a learned function that is generally parameterized as a simple Multilayer Perceptron (MLP).

### 3.2 GRAPH NETWORK SIMULATOR

GNSs simulate a system's dynamics by repeatedly applying the following three steps. First, they encode the system state $\mathcal{S}$ in a graph $\mathcal{G}$. If the system state is given as e.g., a triangular or tetrahedral mesh $\mathcal{M}$ of the underlying entities, this graph is naturally constructed by using the nodes of $\mathcal{M}$ as nodes of the graph, and the connection between these nodes as edges. The node and edge features $\mathbf{X_V}, \mathbf{X_E}$ can be constructed based on the concrete simulation. In general, encoding purely *relative* properties such as relative distances and velocities per edge rather than absolute positions per node have been shown to greatly improve training speed and generalization (Sanchez-Gonzalez et al., 2020). Next, the encoded graph $\mathcal{G}$ is used as input for a learned MPN, which computes final latent representations $x_v^L$ for each node $v \in \mathbf{V}$. These latent representations are interpreted as (potentially higher-order) derivatives of dynamic quantities, which are used by a simple forward-Euler integrator to derive an updated system state $\mathcal{S}'$. Note that for some tasks, only a fraction of mesh nodes need to be predicted, as the others are either fixed or belong to a known entity such as a gripper or collider. In this case, only the latent representations of the nodes with otherwise unknown dynamics are used.

GNSs are trained on a node-wise next-step Mean Squared Error (MSE) objective, i.e., they minimize the 1-step prediction error of the next system state to that of a given ground truth trajectory. During inference, simulations over potentially hundreds of steps can be generated by iteratively repeating the above-mentioned steps, using the updated dynamics of one step as the input for the next. We note that the model does not predict the movement of fixed entities such as e.g., a collider, which is instead assumed to be known and combined with the model's prediction about the unknown parts of the system. Due to this iterative dependence on previous outputs, the model is prone to error accumulation. A common strategy to tackle this limitation is to apply additional noise to the dynamic variables of the system for each training step (Sanchez-Gonzalez et al., 2020; Pfaff et al., 2021). Intuitively, adding training noise acts as a form of data augmentation that allows the learned model to compensate for small prediction errors over time. This kind of error-compensating next-step prediction leads to plausible and visually realistic predictions. However, the resulting predictions can be arbitrarily inaccurate with respect to the true dynamics of the system, since the model has no reference for its simulation other than some potentially incomplete initial state $\mathcal{S}_0$.

## 4 GROUNDING GRAPH NETWORK SIMULATOR

Our approach combines recent advances in graph-based learned physics simulation with additional partial observations of the system state to generate highly accurate simulations from incomplete initial states. To this end, we extend the existing GNS framework to naturally and efficiently integrate auxiliary point cloud data whenever available. This auxiliary information *grounds* the predictions of the model in an observation of the true system state, guiding it towards predictions that not only look realistic but also closely match the actual dynamics of the system. Figure 2 illustrates an overview of our approach. A more detailed description of the GNN-part of the method is found in Appendix A.

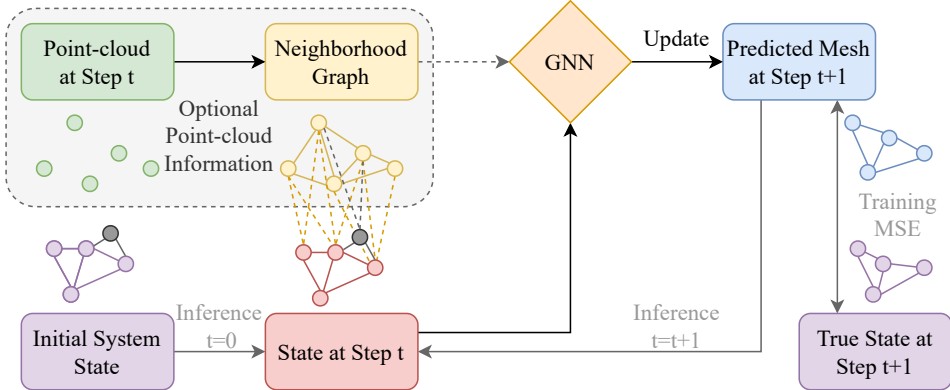

Figure 2: Schematic of GGNS . Given a system state $\mathcal{S}_t$ (red) and optional point cloud observations (dashed box), a GNN (orange) predicts how the system $\mathcal{S}_{t+1}$ will look like at the next step (blue). For object deformation tasks, the state can include boundary conditions (gray) such as colliders or walls. When provided, the point cloud (green) is transformed into a neighborhood graph (yellow) in the same coordinate system as the mesh, connecting each point in the point cloud to the nearest mesh nodes. The model is trained to predict the next system state based on a true state (purple) provided by a ground truth simulator. During inference, the model iteratively predicts updates from a potentially incomplete initial system state (purple), using additional point cloud observations when available.

## 4.1 POINT CLOUDS AND NEIGHBORHOOD GRAPHS

In order to utilize point-based data in addition to meshes we first have to transfer both into a common graph. Following previous work (Sanchez-Gonzalez et al., 2018), we do this by creating a neighborhood graph based on spatial proximity. Given a graph $\mathcal{G} = (\mathbf{V}, \mathbf{E}, \mathbf{X_V}, \mathbf{X_E})$ that encodes a predicted system state and a point cloud observation $\mathcal{P} = \{p_1, \ldots, p_N\}$, $p_j \in \mathbb{R}^d$ of the true system state, we set $\mathbf{V}' = \mathbf{V} \cup \mathcal{P}$ and

$$\mathbf{E}' = \mathbf{E} \cup \{(p_i, p_j) \in \mathcal{P}^2 | d(p_i, p_j) \leq r_{\mathcal{P}}\} \cup \{(v, p), (p, v) | v \in \mathbf{V}, p \in \mathcal{P}, d(p, v) \leq r_{\mathcal{S}}\}.$$

Here, $d$ is some distance measure, usually the euclidean distance, and $r_{\mathcal{P}}$ and $r_{\mathcal{S}}$ are task-specific neighborhood radii. The corresponding features $\mathbf{X_{V'}}$, $\mathbf{X_{E'}}$ of the added nodes and edges in $\mathbf{V}'$ and $\mathbf{E}'$ depend on the concrete task. The different node and edge types are one-hot encoded into their respective features to allow the model to differentiate between them. Similar to the original features, information can be encoded in a relative fashion in the form of edge features to aid generalization. More concretely, we encode relative distances in world space along all edges, additionally adding mesh-space distances for edges between two mesh nodes. This connectivity is slightly different from MGN (Pfaff et al., 2021), which make use of additional *world edges* between mesh-nodes by creating a similar radius-based neighborhood graph for the mesh nodes in world space.

## 4.2 IMPUTATION-BASED TRAINING AND INFERENCE

For most realistic applications, point clouds are typically not available at each time step during inference. For example, we may have access to observed point clouds from the previous $k$ time steps and want to use them to infer the state of the system in the future. We adapt our model to this constraint by employing an imputation-based training scheme. Our model still uses a single GNN, but we now randomly replace the graph $\mathcal{G}$ of $\mathcal{S}$ with the corresponding extended graph $\mathcal{G}' = (\mathbf{V}', \mathbf{E}', \mathbf{X_{V'}}, \mathbf{X_{E'}})$ with equal probability during training. In both cases, the model is only trained to predict the system dynamics for the original nodes $\mathbf{V}$. Intuitively, this allows each system node $v \in \mathbf{V}$ to utilize the additional information of close-by points of a point cloud when available, while at the same time forcing it to also make sensible predictions when there is no additional information. During inference, we construct $\mathcal{G}'$ from the (predicted) system state $\mathcal{S}$ and a corresponding observed point cloud $\mathcal{P}$ of the true object whenever available and use $\mathcal{G}$ otherwise. This enables the model to reason about the true system state that is observed via $\mathcal{P}$, adapting its prediction to the otherwise unknown behavior of the system. This *grounding* of the prediction also alleviates common errors of GNS such as drift and more generally error accumulation. An example can be seen in Figure 1. Here, the system state consists of a predicted mesh and a gripper, and the point cloud consists of points sampled from the true object. The mismatch between point cloud and predicted mesh indicates the prediction error, and

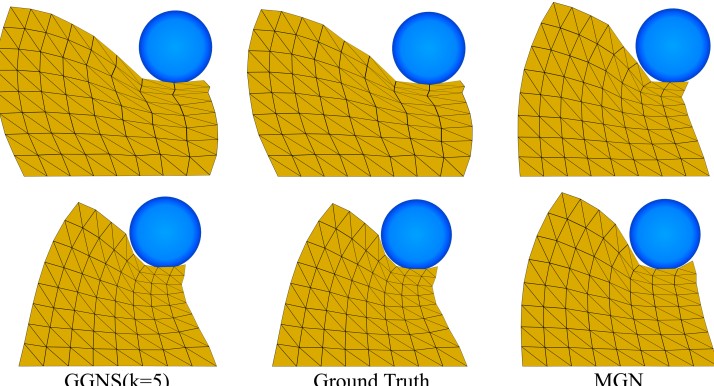

|  |  |  |
|---|---|---|
| GGNS(k=5) | Ground Truth | MGN |

Figure 3: Final simulated meshes ($t = 50$) for GGNS ($k = 5$) (left), the ground truth simulation (middle) and MGN (right) for 2 test rollouts with different material properties for the Deformable Plate task. Our model closely matches the ground truth simulations for both materials, while MGN predicts the same material every time.

the model uses this additional information to correct the current state estimate. Similar figures for the other two tasks can be found in Appendix C.

We compare this simple imputation-based method to another training scheme in our experiments, which we call GGNS+LSTM. Here we use an LSTM (Hochreiter & Schmidhuber, 1997) layer on the node output features of the GNN to explicitly include recurrency into the model. This modification allows information such as the material properties to be inferred and propagated over time, which can be utilized to improve the predictions in time steps without point clouds. The resulting model is trained on the same 1-step prediction loss and also uses training noise to generate stable rollouts during inference. However, it is significantly more costly to train, as it makes use of backpropagation in time to compute the gradients for the recurrency. We find experimentally that this recurrent model performs worse than the imputation-based method. An explanation for this is that the potential benefit of propagating information over time is offset by the additional training and model complexity, especially with respect to the next-step prediction objective. For this reason, GGNS relies on this simple but effective imputation-based approach.

## 5 EXPERIMENTS

We evaluate GGNS on complex 2d and 3d mesh-based object deformation prediction tasks modelled in the Simulation Open Framework Architecture (SOFA) (Faure et al., 2012). For each task, the true system state is given by a tetrahedral FEM mesh of a deformable object with rigid boundary conditions combined with a triangular surface mesh of a rigid collider. The point clouds are generated by raycasting using one virtual camera for $2d$ and up to five cameras for $3d$ tasks arranged around the scene. More details on the generation of the point clouds are presented in Appendix B. Additional environment-specific details, including node and edge features and dataset properties can also be found in Appendix B. We assume that, while the initial mesh of the object is known, its material properties are not. We model these unknown properties via the Poisson's ratio (Lim, 2015) $-1 < \nu < 0.5$, which is a scalar value describing the ratio of contraction ($\nu < 0$) or expansion ($\nu > 0$) under compression (Mazaev et al., 2020). For all datasets, we randomly assign Poisson's ratios from $\nu \in \{-0.9, 0.0, 0.49\}$ equally to all rollouts.

We train all models on all tasks using the Adam optimizer (Kingma & Ba, 2015) with a learning rate of $5 \times 10^{-4}$ and a batch size of 32, using early stopping on a held-out validation set to save the best model iteration for each setting. The models use a LeakyReLU activation function, five message passing blocks with 1-layer MLPs and a latent dimension of 128 for node and edge updates. We use a mean aggregation for the edge features and a training noise of 0.01. All tasks use a normalized task space of $[-1, 1]^d$. An overview of the network hyperparameters can be found in Appendix E.

**Evaluation Metrics.** We evaluate the performance of all trained models on 10 different seeds per experiment. We report the means and standard deviations of the different runs, where, for each run, we average the results over all available steps of a trajectory and over all trajectories in the test set of

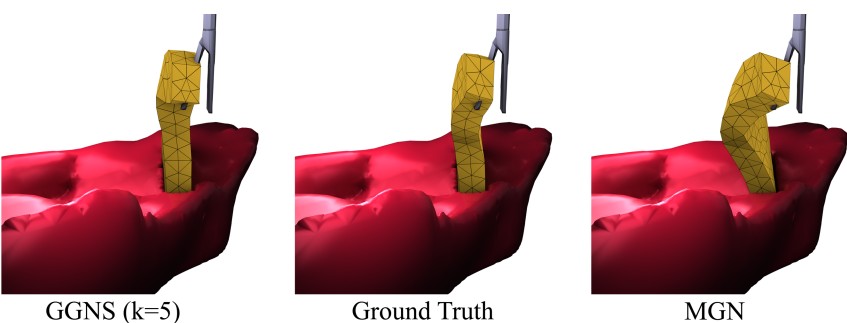

| GGNS (k=5) | Ground Truth | MGN |

Figure 4: Visualization for a test trajectory at time step $t = 70$ for GGNS (left), the ground truth simulation (middle) and MGN (right). While MGN accumulates a large prediction error over time, GGNS is able to utilize the additional point cloud information to stay close to the ground truth for the full length of the simulation. Different angles of this visualization can be found in Appendix C.

the respective data set. For all experiments, we report the *full rollout loss*, where the model starts with the initial state $\mathcal{S}_0$ and predicts the states up to a final state $\mathcal{S}_T$. Here, we provide a point cloud to the model every $k \geq 1$ steps and resort to mesh-only prediction otherwise. This corresponds to a setting in which the deformation of an object is tracked with both high-frequency sensors and low-frequency cameras which provide the position of the rigid collider and point-cloud information respectively.

We also consider an application where a robot observes an object's deformation up to some point in time and then reasons about future deformations without additional point-cloud information. For this setting, the initial system state $\mathcal{S}_0$ is provided to the model, followed by $m$ point clouds for its next $m$ predictions. Then, 10 more steps are predicted without point clouds to predict a state $\mathcal{S}_{m+10}$ and and compute the corresponding 10-*step prediction loss*. The reported losses are the average MSE over every step along the trajectory averaged over all possible rollouts. This metric reduces to the average loss for a $m + 10$-step prediction for methods that cannot make use of point cloud data, as the state $\mathcal{S}_{m+10}$ needs to be predicted from the initial $\mathcal{S}_0$.

**Baselines.** We compare to MGN, a state-of-the-art GNS, which utilizes additional *world edges* between close-by mesh nodes, but does not incorporate point cloud observations. Comparing these world edges to Section 4.1, MGN assumes an edge partition $\mathbf{E} = \mathbf{E}_1 \dot{\cup} \mathbf{E}_2$ and separate edge update functions $f^l_{\mathbf{E}_1}$ and $f^l_{\mathbf{E}_2}$. The edge-aggregation for the node update is then computed by aggregating the latent features of both types of edges separately and concatenating the result. We adopt this explicit representation of edge types for the MGN baseline and experiment with it for GGNS in Appendix D. As it does not provide any significant advantages for our model, GGNS instead resorts to a simple one-hot encoding of the type of input edge for the remaining experiments.

Additionally, we evaluate a variant of MGN that has additional access to the underlying Poisson's ratio $v$ as a node feature, called *MGN (M)*. This additional information leads to a deterministic ground truth simulation w.r.t. the initial system state, and upper bounds the performance of MGN. We also compare to GGNS+LSTM, which integrates recurrency into our imputation technique. Here, we investigate whether this recurrency helps the model predicting e.g., material properties over time.

As a point cloud based baseline, we use a non-learned method to directly generate a mesh from the point cloud of each time step. We voxel-subsample the point cloud so that we observe approximately the same number of points as nodes in the ground truth mesh and then use *Alpha Shapes* (Akkiraju et al., 1995) to create a (potentially non-convex) mesh for this time step. This baseline shows how much information can be directly inferred from just the point cloud information.

**Deformable Plate.** We consider a family of 2-dimensional trapezoids that are deformed by a circular collider with constant velocity. Besides the trapezoidal shapes, diversity in the dataset is introduced by varying the size and starting positions of the collider. For this task, we additionally consider the Intersection over Union (IoU) between the predicted and the ground truth mesh as an evaluation metric. We find that this metric is less sensitive to individual mesh nodes and that it instead measures how well the predicted object shape matches that of the real system state. We use a total of $675/135/135$ trajectories for our training, validation and test sets. Each trajectory consist of $T = 50$ timesteps.

**Tissue Manipulation.** An important application for the prediction of deformable objects is medical robotics. We simulate a robot-assisted surgery scenario where a piece of tissue is deformed by a

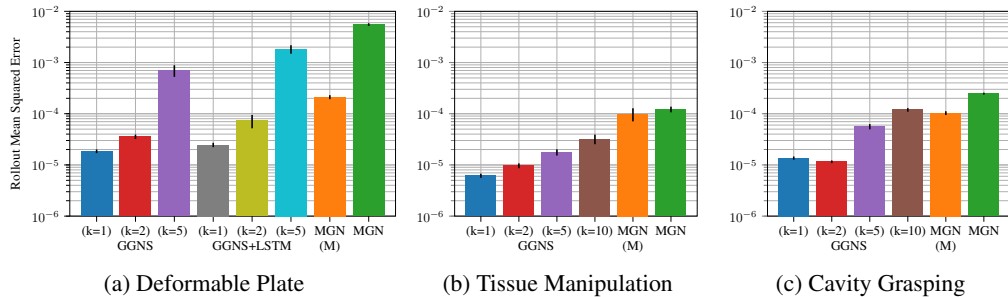

(a) Deformable Plate       (b) Tissue Manipulation       (c) Cavity Grasping

Figure 5: Rollout Mean Squared Error of GGNS and MGN baselines evaluated on the test set of the three datasets. We report the results for GGNS using point clouds in every $k$-th time step. MGN(M) indicates the baseline method of MGN that uses the ground truth material as input feature. GGNS outperforms the MGN baseline in all settings and in most cases even if it has access to the complete initial state. For the Deformable Plate task we additionally report the errors for GGNS+LSTM, which perform worse than GGNS for all $k$ but still outperforming the MGN baseline.

solid gripper. Varying the direction of the gripper's motion and its gripping position on the tissue results in additional diversity. Here, $600/120/120$ trajectories are used, each of which is rolled out for $T = 100$ timesteps. This task is visualized in Figure 4.

**Cavity Grasping.** Robotic manipulation of deformable objects is an important application of deformable physics simulation. Here, a simulated Panda [1] robot gripper grasps and deforms a cavity. For this purpose, we randomly generate cone-shaped cavities with different radii, which are deformed by a gripper from different positions. An example simulation step by GGNS for this task is illustrated in Figure 1. We use the same amount of samples and data split as in the Tissue Manipulation task.

## 6 RESULTS

**Main Results.** We test our method on the three deformation prediction tasks described in Section 5 and compare it to MGN with and without material information. We find that GGNS can use the point cloud information to produce high quality rollouts that closely match the true system states. An example is shown in Figure 3, which visualizes the final simulated meshes for our method and the ground truth simulation. Additionally, GGNS outperforms the baselines even when they have access to the complete initial state, which our model has not. Figure 4 shows the qualitative differences between GGNS and MGN on the Tissue Manipulation task. Additional visualizations for all tasks and both methods can be found in Appendix C. The evaluations for full rollouts are given in Figure 5. Table 1 shows results for the $m + 10$-step evaluation. Appendix D shows the performance of GGNS for different model hyperparameters. Similar to Pfaff et al. (2021), we find that GGNS is robust to most parameter choices, and that a modest amount of training noise is crucial for long-term rollouts. To show the applicability of our method for more realistic point cloud data, we provide additional ablations on noisy and partial observable point clouds in Appendix D. We find that our model is quite robust to the quality of the point clouds and can still reliably use their information to ground the simulation. On the Deformable Plate dataset, we additionally evaluate the mean Intersection over Union (IoU) during the rollouts to emphasize the compliance with the overall shape of the object rather than that of individual mesh nodes. The results are illustrated in Figure 6a.

**Recurrent Imputation Model.** For the $2d$ data of the Deformable Plate task, we additionally compare our imputation model to the GGNS+LSTM approach, which can use the recurrence of LSTMs to pass information over time. Figure 5a shows that GGNS outperforms this alternative approach for each $k$. We find that our simple architecture outperforms the recurrent one while requiring significantly less time to train, likely due to the additional complexity of training the recurrent model. The qualitative results in Appendix C confirm these findings.

**Initial Mesh Generation.** Using the IoU metric, we can compare objects across different mesh representations. The results in Figure 6b show that GGNS produces accurate rollouts even if the initial mesh is generated directly from the initial point cloud. For this, we compute a mesh with similar resolution to the training meshes from the convex hull of the initial point cloud, avoiding

---

[1]FRANKA EMIKA GmbH, Munich, Germany

| Approach | $m + 10$-step MSE $\times 10^{-5}$ | | |
| --- | --- | --- | --- |
| | Plate | Tissue | Cavity |
| GGNS | $2.907 \pm 0.172$ | $0.514 \pm 0.052$ | $0.923 \pm 0.040$ |
| MGN (M) | $10.663 \pm 1.063$ | $5.027 \pm 1.489$ | $6.294 \pm 0.668$ |
| MGN | $282.684 \pm 18.112$ | $5.885 \pm 0.723$ | $11.528 \pm 0.747$ |

Table 1: Evaluation on the $m + 10$-step prediction setting on the test set for all three tasks. GGNS clearly outperforms the baselines on all tasks even if they have access to the full initial simulation state.

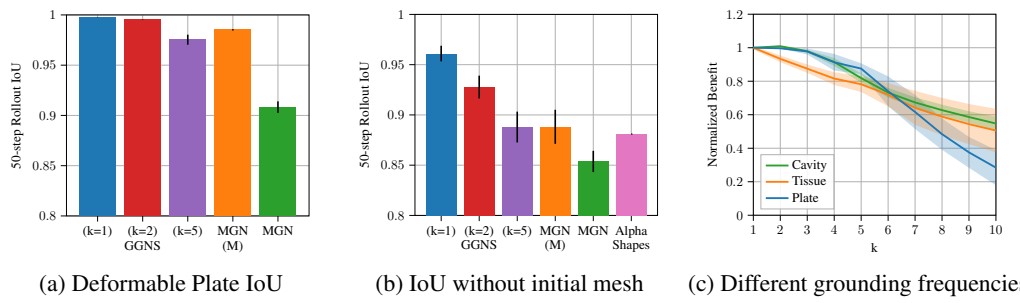

(a) Deformable Plate IoU    (b) IoU without initial mesh    (c) Different grounding frequencies

Figure 6: (a) Rollout IoU for the Deformable Plate dataset for GGNS and the MGN baseline. The main findings here are similiar to the MSE results but enable to compare against the values in (b), where the initial mesh is created from initial the point cloud. The achieved IoUs are lower compared to using the ground truth mesh, but GGNS ($k \leq 2$) still outperforms all baselines including the Alpha Shapes in this setting. (c) Normalized benefit of using a point cloud in every $k$-th timestep, where $k = 1$ means a point cloud is available in every time step.

the dependence on *any* simulation data. This procedure marks an important step towards using these models on real world data. The results indicate that generating the initial mesh from point cloud information results in a degradation of the performance compared to an evaluation that uses a provided mesh. Yet, it still allows for a high-quality prediction of the deformation. The comparison to Alpha Shapes shows that combining infrequent point cloud information ($k = 5$) with a simulator leads to better and more consistent results than directly creating the mesh from the point cloud in each time step. Additionally, our model naturally tracks the correspondences of mesh nodes over time, whereas Alpha Shapes cannot observe the evolution of individual particles in the system. As such, GGNS allows for a more thorough understanding of the modeled process.

**Grounding Frequency.** Figure 6c shows the normalized performance of GGNS for grounding frequencies $k \in \{1..10\}$ across tasks. Here, a value of $1.0$ corresponds to the performance for $k = 1$, and $0.0$ to the performance MGN. For all tasks there is a clear advantage in utilizing the point cloud information, and the performance increases with the frequency of available point clouds.

## 7 CONCLUSION

We propose Grounding Graph Network Simulator (GGNS), an extension of the popular Graph Network Simulator framework that can utilize auxiliary observations to accurately simulate complex dynamics from incomplete initial system states. Utilizing a neighborhood graph computed from point cloud information and an imputation-based training scheme, our model is able to *ground* its prediction in an observation of the true system state. We show experimentally that this leads to high-quality simulations in challenging $2d$ and $3d$ object deformation tasks, outperforming existing approaches even when these are provided with full information about the system.

In future work, we will extend GGNSs to explicitly model uncertainty and maintain a belief over the latent variables of the system, e.g., by employing a Kalman filter in a learned latent space (Becker et al., 2019). Another promising direction is to adapt the current next-step prediction loss to instead predict a trajectory over a small period of time to increase the long-term consistency of the model. Finally, we will employ our model for model-predictive control and model-based Reinforcement Learning in both simulation and on a real robot.

ACKNOWLEDGMENTS

We thank Vincent Kreuziger for the helpful discussions on the visualizations and for the high-quality blender renderings. The authors acknowledge support by the state of Baden-Württemberg through bwHPC. GN was supported by the DFG research unit DFG-FOR 5339 (AI-based Methodology for the Fast Maturation of Immature Manufacturing Processes) and GN and NF were supported by the BMBF project Davis (Datengetriebene Vernetzung für die ingenieurtechnische Simulation).

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

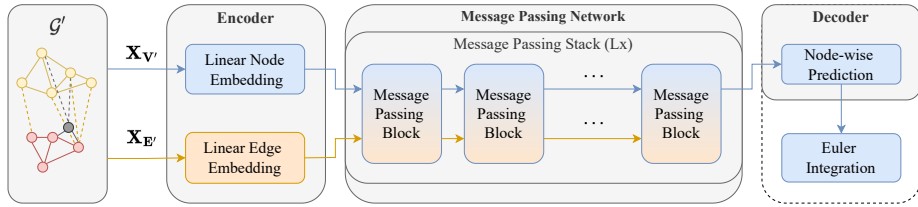

Figure 7: A detailed view of the GNN part of GGNS . Given a graph $\mathcal{G}'$, the node and edge features $\mathbf{X}_{\mathbf{V}'}$ and $\mathbf{X}_{\mathbf{E}'}$ are linearly embedded into a latent space and then updated with $L$ Message Passing Blocks. The resulting predictions are interpreted as dynamic quantities that are used to update the system.

## A  MODEL DETAILS

The Message Passing Network employed by GGNS is displayed in 7. As node-wise predictions we use velocities, which are Euler-integrated once to update the positions of the mesh of the deformable object.

## B  ENVIRONMENT DETAILS

Here, we describe all key aspects, which are valid for all three environments. All datasets are simulated using SOFA and include different material properties. Therefore, we choose discrete Poisson's ratios from $\nu \in \{-0.9, 0.0, 0.49\}$ for one-third of all simulated trajectories each. Other material parameters are kept constant, e.g., for the mass we choose large values for the solid object and smaller values for the deformable to ensure sufficient deformation. The chosen parameters do not represent the full reality, as there are other material parameters that could be varied. However, as we want to showcase the capabilities of our method, we selected these parameters as they displayed the biggest impact on the deformation behavior.

### B.1  POINT CLOUD GENERATION

The required point clouds are not directly available in SOFA, but instead rendered from the scene of the meshes using *Raycasting* from Open3D (Zhou et al., 2018). We therefore place virtual cameras around and on top of the scene to generate partial point clouds from different directions. For the Deformable Plate dataset one camera is sufficient, while the other two tasks rely on four cameras around and one camera on top of the scene. This results in a good, but not complete coverage of the entire surface with points of the point cloud. Even though there are five cameras around the scene, there are areas that are not covered: For the tissue, the parts that are occluded by the red liver, and for the cavity, parts of the inner surface depending on how the upper and lower radii deviates from one another. Also, as there can be no camera from below, there are naturally no points on the lower surface for both datasets. In Appendix D we additionally provide results for less cameras on the cavity dataset, leading to only partially observable point clouds. If more than one point cloud camera is used, the resulting point clouds are fused and subsampled accordingly to achieve a processable number of points. We voxel subsample in world space, so the points do not belong to any specific part of the mesh, but can rather be seen as some "interpolation" between mesh vertexes. The main challenge is that there are no point correspondences and that the model needs to figure out which point of the point cloud belongs to which vertex in the mesh to do the correction of the mesh nodes for grounding the simulation. Still, voxel subsampling leads to the most structured results compared to other subsampling techniques, which helps the model to account for correspondences between points over time.

## B.2 INPUT FEATURES

In addition to encoding the node or edge type as one-hot features, we add an encoding to static nodes and encode the velocity of the collider in its node features. We encode the positions in space as relative features in the edges instead of absolute encodings in the node features following previous work (Sanchez-Gonzalez et al., 2020). All edges thus receive their relative world coordinates, while mesh edges additionally contain relative coordinates in mesh space.

## B.3 COLLISION HANDLING

SOFA as the ground truth simulator handles collision between objects using triangular surface meshes of all objects involved to detect collisions. The detection is implemented using the *LocalMinDistance* method and detected collisions are included in the constraints of the system. Using Lagrangian multipliers, the constraints are then processed together with the other forces from the deformation to solve the complete FEM system (Faure et al., 2012). In contrast to that, GGNS uses one-hot encoded edges between the rigid and the soft body that are used by the model to compute the dynamics. There is no explicit handling of collisions, the network learns to avoid them and adapts the mesh accordingly.

## B.4 DEFORMABLE PLATE

For this environment, we simulate a family of 2-dimensional trapezoids deformed by a circular collider with constant velocity. We vary the size of the collider by sampling from a triangular distribution between 15 and 60 % of the edge length of the deformable object. For the collider start position we sample from a uniform distribution between the left and right corner of deformable object. We record 50 time steps per trajectory and 945 trajectories in total, which are split in $675/135/135$ trajectories per train, evaluation and test set. A single data sample contains approx. 700 nodes: 57 nodes for the collider, 81 nodes for the mesh oft the deformable object and around 600 points in the subsampled point cloud. The mesh itself consists of 416 edges, the total number of edges is about 3 K depending on the deformation in the according time step. In contrast to the Poisson's ratio, the other adjustable material parameter in SOFA, the Young's modulus is kept constant for all samples at $E = 5\,000$Pa. It describes the compressive stiffness when a force is applied lengthwise. The different material properties together with the different trapezoidal shapes introduce uncertainty in the form of multi-modality into the data. The reason for this is that different deformations result in states that cannot be clearly assigned to a single trapez-material combination. We construct this dataset because it comes with lower computational cost due to the restriction to 2d, but already allows for more general statements due to the non-trivial deformations and the multi-modality. Therefore, it is especially suitable as a proof-of-concept and for ablations.

## B.5 TISSUE MANIPULATION

Here, a piece of tissue is deformed by a rigid gripper which could be part of a robot-assisted surgery scenario. To generate diversity, we generate random motions in a $2d$ plane and sample a random gripping point from the 19 top mesh points. We record 100 time steps per trajectory and 840 trajectories in total, which we split in $600/120/120$ trajectories per train, evaluation and test set. A single data sample consists of approx. $1\,200$ nodes: 361 for the mesh, one for the gripper and about 850 for the point cloud. The mesh consists of $2\,154$ edges, which leads to a total number of about $3\,800$ edges depending on the time step. To ensure physically plausible deformation, each Poisson's ratio is assigned its specific Young's modulus from $E \in \{10\,000, 80\,000, 30\,000\}$Pa. If instead it were kept the same for each Poisson's ratio, the gripper could penetrate the deformable object or pull it without touching it. The uncertainty in this dataset is mainly in the initial state, which can result in different deformations depending on the material from the same initial state.

## B.6 CAVITY GRASPING

We randomly generate cone-shaped cavities with radii between $87.5\%$ and $50\%$ of the maximum possible gripping width. The cone shape helps to increase uncertainty in the form of multi-modality in the data, because the states resulting from deformation cannot be clearly assigned to a single

cone-material combination. The deformable cavities are deformed by a gripper located at random positions in space. The positions are sampled form a hexahedron around the geometrical center of the cavity ensuring collision free starting positions. For the grasping, the gripper moves as quickly as it is allowed to the gripping position and then closes its fingers with constant velocity. We record 100 time steps per trajectory and 840 trajectories in total, which are split in 600/120/120 trajectories per train, evaluation and test set. A single data sample consists of approx. 2.4 K nodes: 750 for the mesh, 636 for the gripper and about 1 K for the point cloud. The mesh consists of 4 500 edges, the overall number of edges in the graph is about 8.5 K depending on the exact time step. The motivation for the creation of this environment is that a successful use of our method in this setting is an important step on the way to a real-world application.

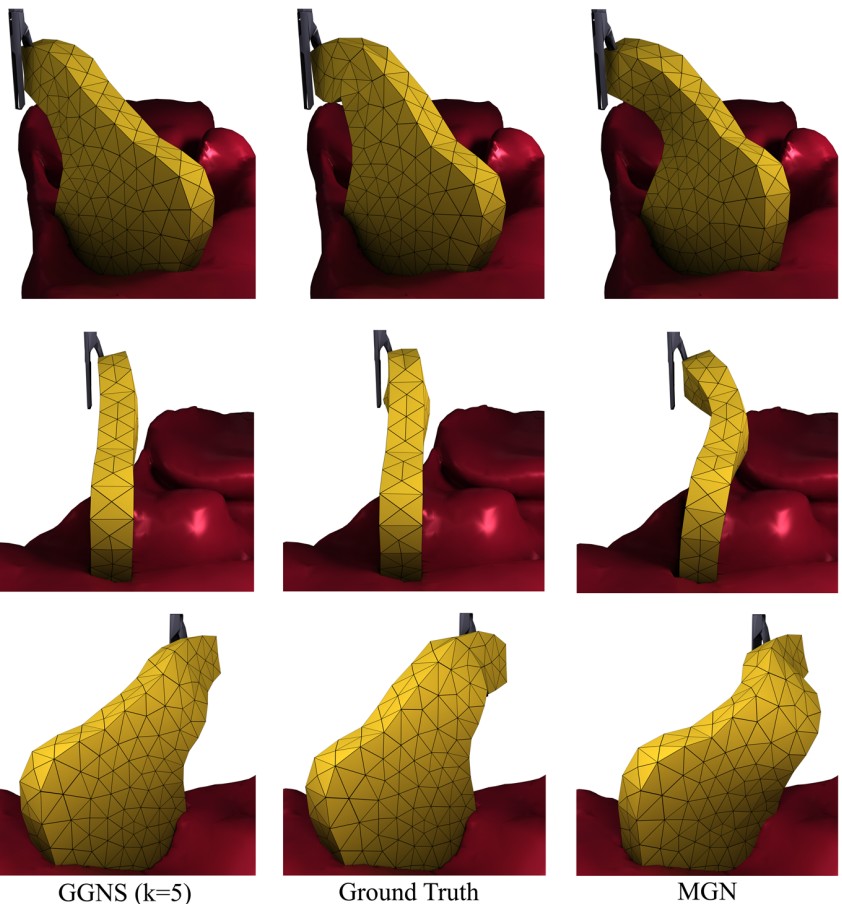

GGNS (k=5)        Ground Truth        MGN

Figure 8: Visualization of a test trajectory in the Tissue Manipulation dataset from three different viewing angles (rows) at time step $t = 70$ for GGNS (left), the ground truth simulation (middle) and MGN (right).

## C    QUALITATIVE RESULTS

In addition to the qualitative illustrations in the main paper, we also provide further views and examples here: Figure 8 shows the same trajectory as Figure 4 but from three additional viewing angles. Figure 9 and Figure 10 show an overlay of the point cloud on the deformable object during the time step where the simulation is grounded by the point cloud. This representation is comparable to Figure 1 for the Cavity Grasping dataset. Furthermore, we provide example visualizations for a test rollout over time for the Deformable Plate task in Figure 11, for the Tissue Manipulation task in Figure 12, and for the Cavity Grasping in Figure 13. Throughout all tasks, GGNS closely matches the ground truth simulation for the complete rollout, achieving close to optimal results when provided with frequent point cloud information ($k = 2$). Opposed to this, MGN sometimes fails to predict the correct material, leading to poor predictions over time and large mismatches in the final system states.

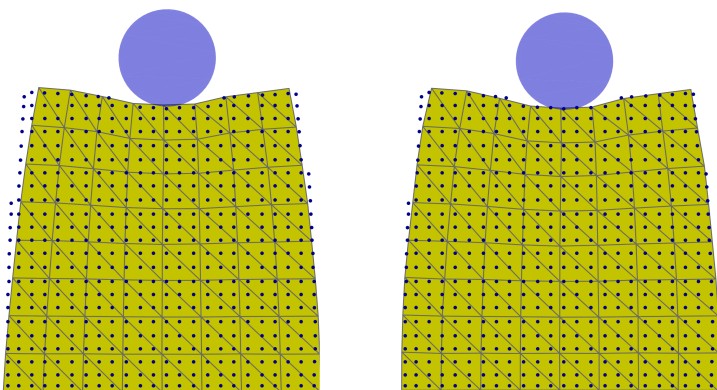

Figure 9: Overlay of the point cloud and the predicted mesh for two consecutive time steps $t = [10, 11]$ in the Deformable Plate dataset. We repeat the point cloud from the earlier simulation step in both images for clarity. The illustration shows the correction behavior of GGNS by including the point cloud to ground the mesh based simulation in this time step. This can be observed particularly well in the upper left and right corners of the plate.

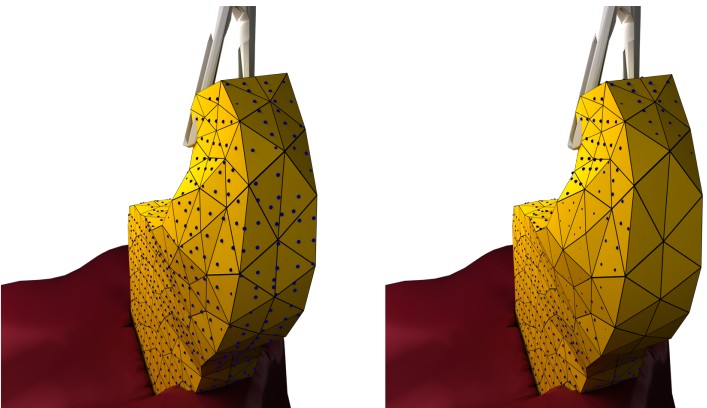

Figure 10: Overlay of the point cloud and the predicted mesh for two consecutive time steps $t = [70, 71]$ in the Tissue Manipulation dataset. We repeat the point cloud from the earlier simulation step in both images for clarity. The illustration shows the correction behavior of GGNS by including the point cloud to ground the mesh based simulation in the time step.

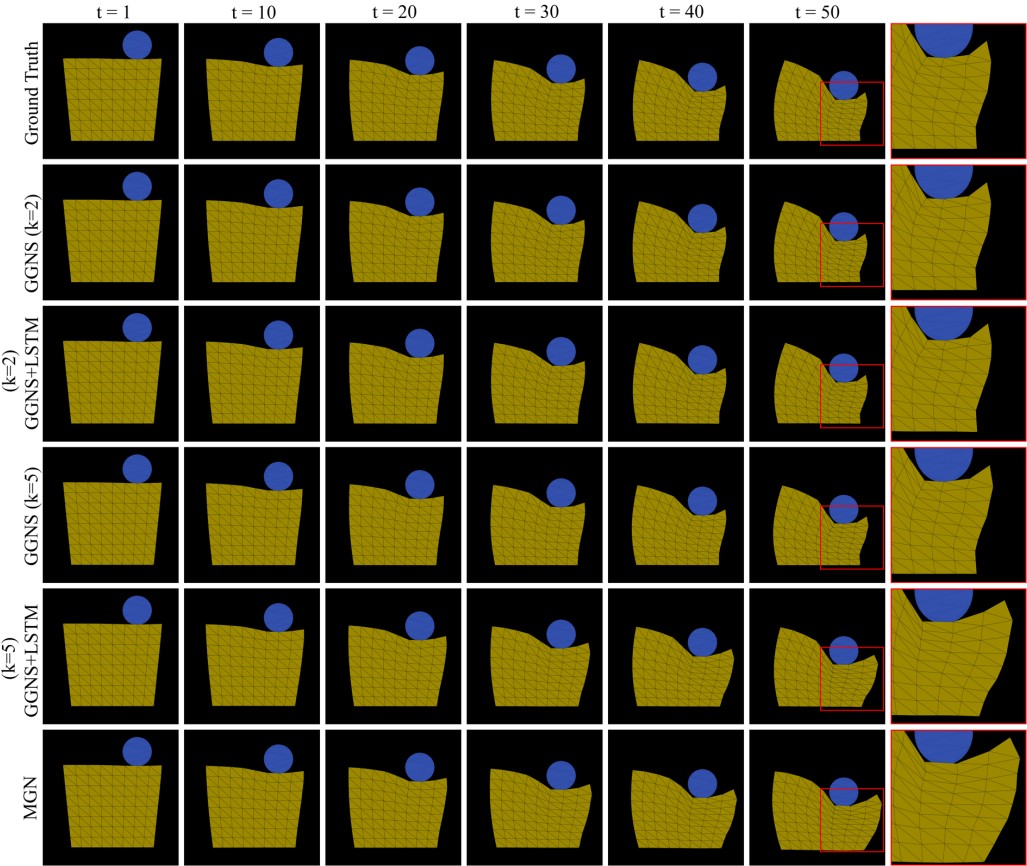

Figure 11: Test rollout visualization for the Deformable Plate task. The last column depicts a close-up of the final time step, which is shown in full in the previous column. Here, we additionally show qualitative results for the GGNS+LSTM model. We can see that for $k = 2$ it matches the ground truth quite well, while for $k = 5$ a large error occurs due to a prediction of the wrong material.

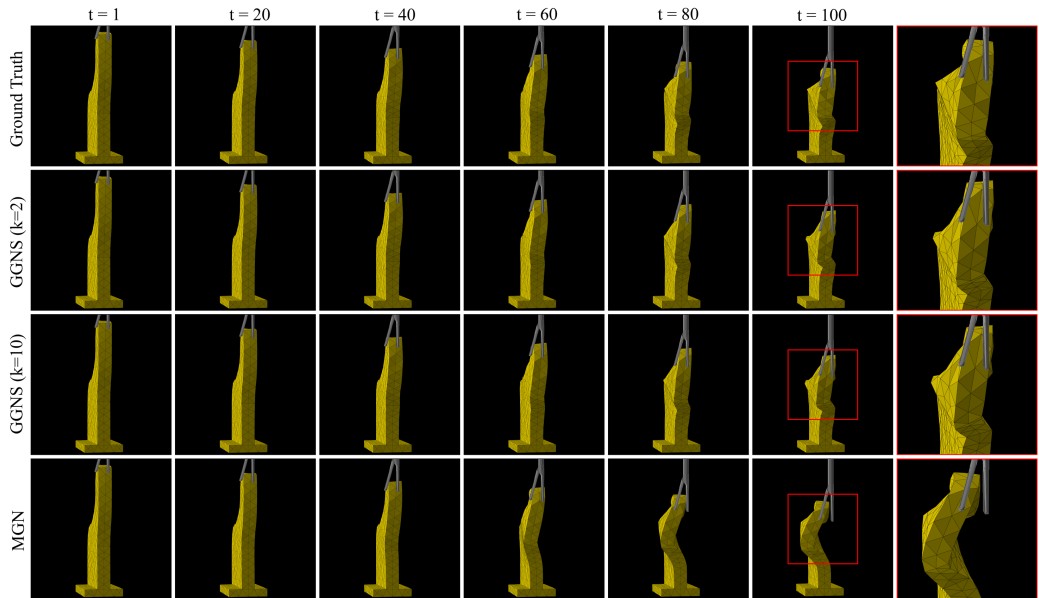

Figure 12: Test rollout visualization for the Tissue Manipulation task. The last column depicts a close-up of the final time step, which is shown in full in the previous column.

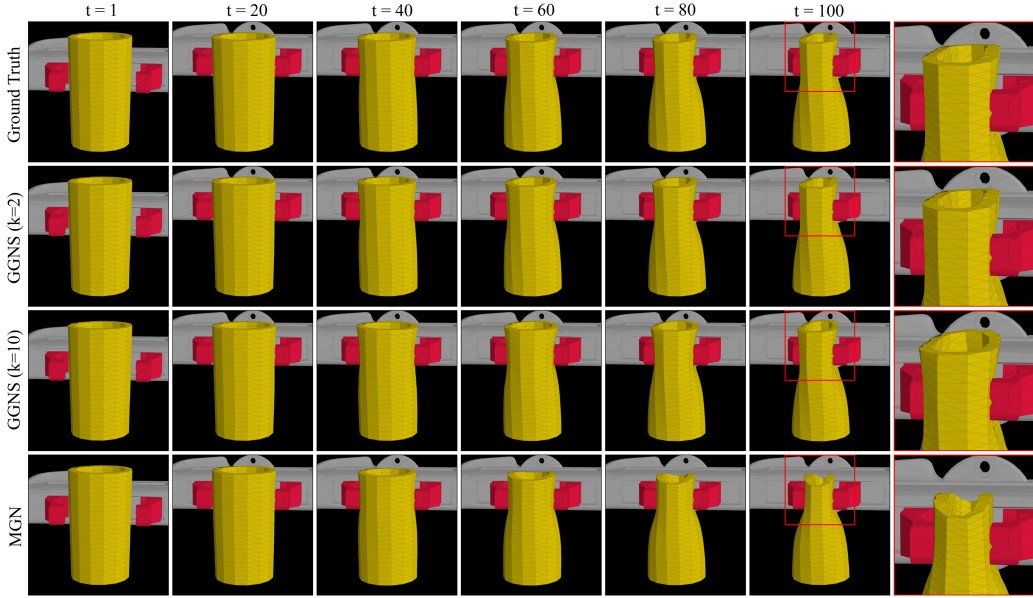

Figure 13: Test rollout visualization for the Cavity Grasping task. The last column depicts a close-up of the final time step, which is shown in full in the previous column.

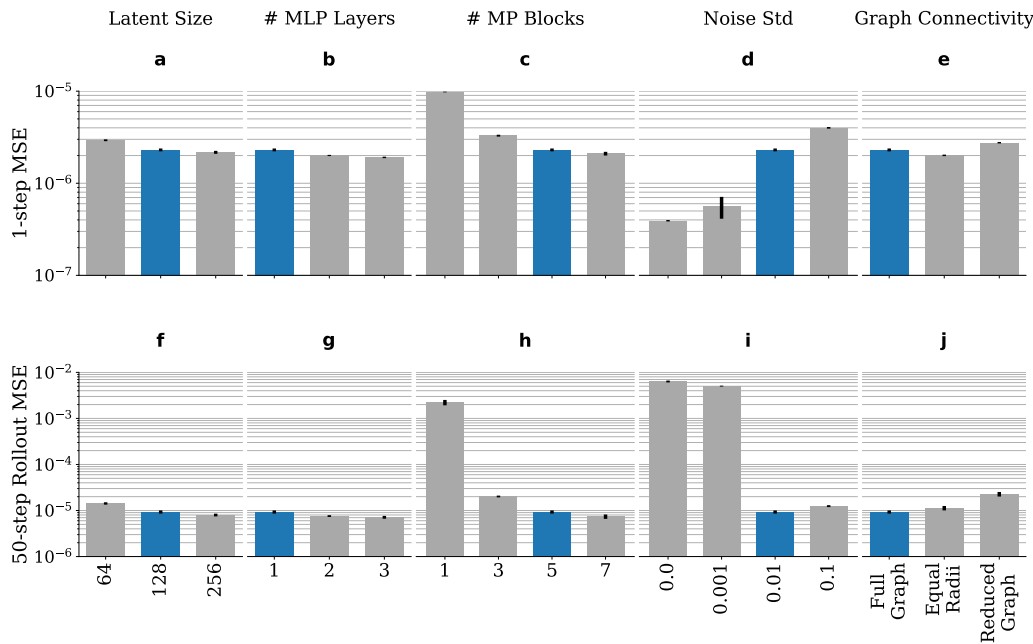

Figure 14: Performance for different changes in hyperparameter choices (grey) on the Deformable Plate dataset in comparison to our default model (blue) with $k = 1$. Error bars indicate one standard deviation. The top row shows the error for the next-step prediction, the bottom row that of full rollouts. We find that a suitable noise scale is crucial for stable rollouts, and that more information in the form of additional edges between the different types of graph nodes generally improves performance. Given enough Message Passing (MP) blocks, further increases in model capacity only lead to modest improvements.

# D   ABLATIONS

## D.1   HYPERPARAMETER CHOICES

Figure 14 compares the performance of GGNS for different hyperparameter choices. We find that the most importance parameters are the number of Message Passing (MP) blocks and the scale of the noise used in training. Both are crucial to achieve a good performance over multi-step rollouts. In terms of training noise, there is a 1-step/multi-step loss trade-off. Other than that, our approach is robust to variations of the different hyperparameters. In terms of graph connectivity, it can be seen that all settings achieve similar performance. Additional information in the form of more local edges helps slightly, while larger connectivity radii do not do much. A detailed listing of the used edge radii is display in Table 2. In particular, the use of significantly more edges in the *Equal Radii* setting does not provide a significant advantage, which is why we use weaker connectivity *Full Graph* that saves computation time. The results for the *Reduced Graph* settings show that edges within the point cloud are not mandatory. For this reason, we omit these edges in the more complex $3d$ tasks in favor of shorter computation time.

## D.2   NOISY POINT CLOUDS

Besides the ablations on our hyperparameter choices, we present further ablations on more realistic point cloud data. For this purpose, we use point clouds with additional noise and only partial observability to get closer to real world point clouds. Figure 15 shows the results for additional ablations on different scales of noise on the point cloud data of the Deformable Plate dataset. We add noise to the point cloud positions during training, evaluation and testing. This makes it more difficult to infer the correct behavior from the point cloud, but provides a more realistic scenario for, because real world point clouds often exhibit large noise. The results show the robustness of our

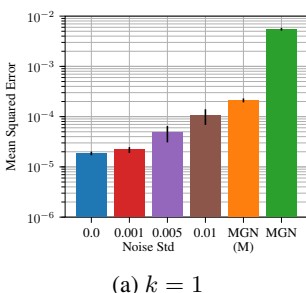 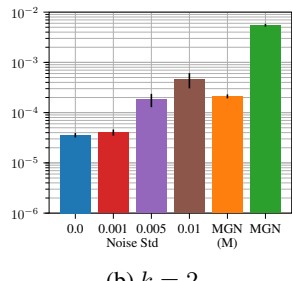 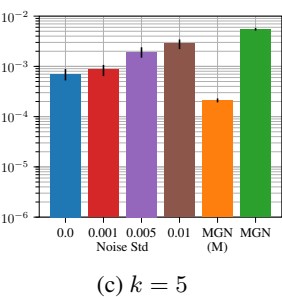

(a) $k = 1$        (b) $k = 2$        (c) $k = 5$

Figure 15: Additional ablations for more realistic point cloud data on two datasets. Here, four different noise levels on the point cloud are evaluated on the Deformable Plate datset. Different grounding frequencies of $k = 1$ in (a), $k = 2$ in (b) and $k = 5$ in (c). GGNS performs better than the baseline even when noise in the scale of the training noise of $\sigma = 0.01$ is applied to the point cloud.

method: Even when a noise level of $\sigma = 0.01$ is applied to the point cloud during testing, it clearly outperforms the baseline. This noise level corresponds to the amount of noise used on the mesh during training.

### D.3 PARTIAL OBSERVABLE POINT CLOUDS

For the ablations on the partial observability, we use the Cavity Grasping dataset. We generate the partial point clouds by using only one, two or five virtual point cloud cameras when using raycasting. The resulting point clouds are visualized for better clarity in Figure 17 for an example test trajectory at time step $t = 0$. One camera results in a coverage from only one half of the outer surface of the cavity and two cameras cover almost the complete outer hull but not the inner surface. With five cameras, the point cloud covers almost the entire mesh completely, except for the inside and bottom. The resulting point clouds have a very different number of points: About $400$ for one camera, about $600$ for two cameras, and about $1000$ for five cameras compared to $750$ mesh nodes for the cavity. The results in Figure 16 show that even with these much less complete point clouds, GGNS still outperforms the baseline. For $k \leq 5$ this is the case even if the baseline has access to the full initial state, which GGNS has not.

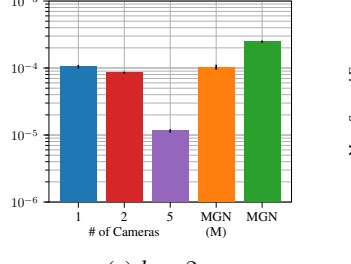 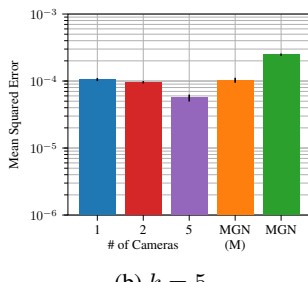 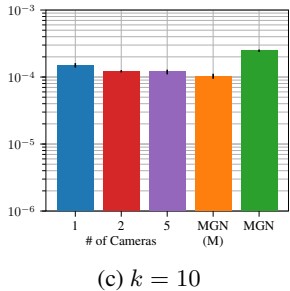

(a) $k = 2$      (b) $k = 5$      (c) $k = 10$

Figure 16: Additional ablations for more realistic point cloud data on the Cavity Grasping dataset. For this purpose, different numbers of cameras are used when generating the point cloud using raycasting. Comparison for three different grounding frequencies: $k = 2$ in (a), $k = 5$ in (b) and $k = 10$ in (c). GGNS outperforms the baseline for all camera settings and grounding frequencies $k$.

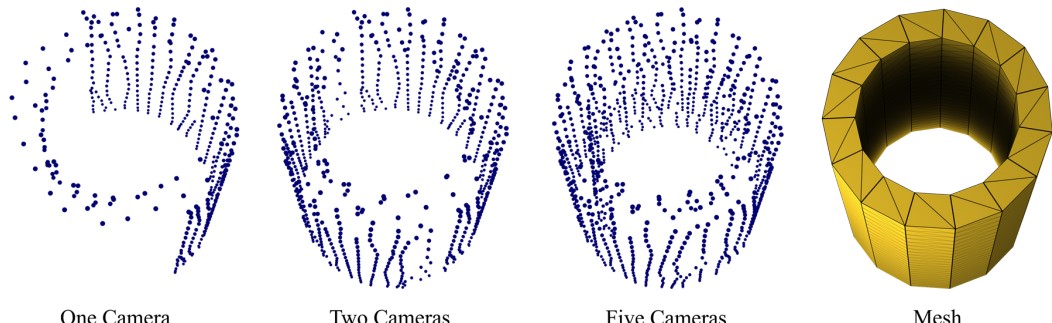

One Camera      Two Cameras      Five Cameras      Mesh

Figure 17: Visualization of the point clouds using one, two or five cameras for the raycasting and the corresponding mesh for reference. It is clearly visible how better coverage of the object is achieved as the number of cameras increases.

Table 2: Edge radii for the connectivities between point clouds $\mathcal{P}$ and meshes $\mathcal{M}$ on the 2D Deformable Plate Dataset.

| Setting | $\mathcal{P} - \mathcal{P}$ | $\mathcal{M} - \mathcal{P}$ | World |
|---|---|---|---|
| Full Graph | 0.1 | 0.08 | - |
| Equal Radii | 0.2 | 0.2 | - |
| Reduced Graph | 0.0 | 0.08 | - |
| MGN | 0.0 | 0.0 | 0.35 |

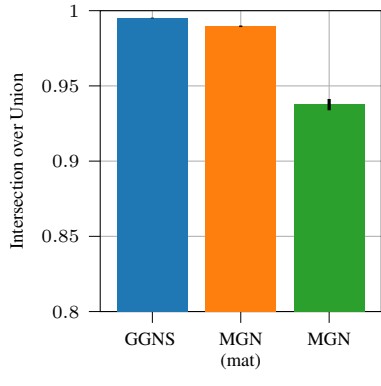

(a) $m + 10$-step IoU Evaluation

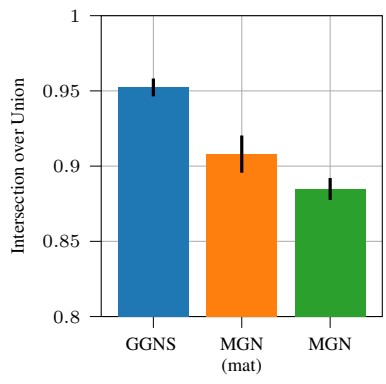

(b) $m + 10$-step IoU Evaluation without initial mesh

Figure 18: (a) Comparison of our model to the baseline results on the Plate cataset using the $m + 10$-step Evaluation routine. (b) Results when using an initial mesh generated from the point cloud. GGNS outperforms the MGN baseline even if it has access to the initial ground truth mesh.

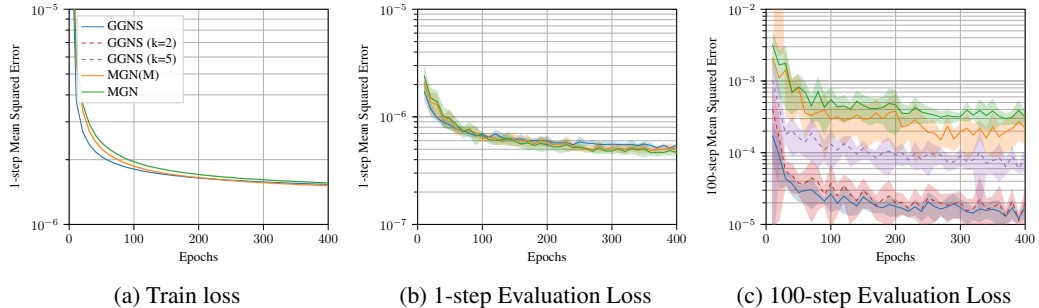

(a) Train loss

(b) 1-step Evaluation Loss

(c) 100-step Evaluation Loss

Figure 19: Exemplary learning curves for the Cavity Grasping task. The light shaded area indicates one standard deviation. Both GGNS and the baselines learn the task pretty similarly in terms of 1-step predictions. Our model is only evaluated for the $k = 2$ and $k = 5$ variant during full rollout evaluation. Here, we can clearly see the advantage of using the point cloud information.

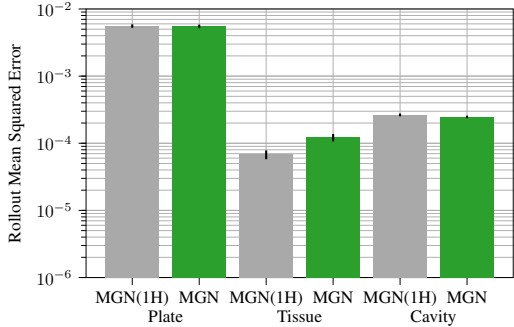

Figure 20: Comparison of the MGN baseline with a version using the one-hot encoded edge types instead of an explicit edge type partitioning indicated by *MGN (1H)*. Both are compared for all three tasks and no significant advantage of the explicit edges partitioning could be found. For this reason, GGNS uses the one-hot encoding, because it is both conceptually simpler and requires less computational power. The MGN baseline still uses explicit edge type partitioning throughout this work, following Pfaff et al. (2021).

Table 3: Configuration of the hyperparameters and key information of the training of our model for all experiments.

| Parameter | Value |
| --- | --- |
| Batch Size | 32 |
| Optimizer | Adam |
| Learning Rate | $5 \times 10^{-4}$ |
| Activation Function | LeakyReLU |
| Aggregation Function | Mean |
| Encoder | Linear Layer |
| MP-Blocks | 5 |
| MLP Layers | 1 |
| Latent Dimension | 128 |
| Decoder | 1-layer MLP |
| Residuals Connections | Around each MP block |
| Training Noise Std | 0.01 |

Table 4: Task specific configuration and hyperparameters for our experiments. We vary the graph connectivity and the number of training epochs for different tasks to control the total training time of our method.

| Parameter | Plate | Tissue | Cavity |
| --- | --- | --- | --- |
| Connectivity Setting | Full Graph | Reduced | Reduced |
| Number of Epochs | 1000 | 800 | 400 |
| Approx. Training Time | $21 : 00$ h | $40 : 00$ h | $38 : 00$ h |

## E  HYPERPARAMETERS

Table 3 gives an overview of hyperparameters shared across tasks. Since GNS are generally robust to the choice of hyperparameters (c.f. D), we use the same hyperparameters for all task and for both, GGNS and MGN for simplicity. The only hyperparameters that vary over tasks are the graph connectivity and the number of training epochs, as shown in Table 4. We adapt these parameters to control for the total training time on a single GPU.

