# OpenReview forum: "Grounding Graph Network Simulators using Physical Sensor Observations"
_ICLR.cc/2023/Conference — ICLR 2023 poster_

### Official Review · Reviewer_ZGJG · 2022-10-21

**Confidence:** 3
**Correctness:** 3
**Technical Novelty And Significance:** 2
**Empirical Novelty And Significance:** 3
**Recommendation:** 6

**Clarity, Quality, Novelty And Reproducibility:**

I think the paper is easy to follow with most of the low-level details properly left out in the supplemental materials.

Fig. 1 was a bit confusing when I read the paper for the first time, but most of it became understandable after reading the corresponding sections:
- It is still unclear to me what solid and dashed lines in Fig. 1 attempt to imply, although it is not a very big deal.
- Why is there an inference arrow pointing from GNN outputs back to “State at Step t”?

The technical method does not seem complicated to implement. Therefore, I think the paper is probably reproducible. It could be better if the authors could release the code.

One thing that is unclear to me is how the collisions between the rigid and soft bodies are handled in their experiments, and it would be nice if the supplemental materials could include more detailed descriptions about it.


**Strength And Weaknesses:**

**Strength**
The paper tackles a critical research problem: error accumulation from long-term prediction of graph neural simulators. Fusing sensor data into dynamic prediction is a reasonable strategy. The paper is also well-written and articulates its core idea clearly.

**Weakness**
Regarding the technical method:
I think using point cloud to enhance dynamic prediction is a good direction to explore, but I feel the proposed method is not quite ready and the problem setup in this paper is a bit contrived: the point cloud data is generated from the ground truth mesh in a simulator and seems much denser than the mesh nodes (the DoFs in simulation). I would be much more excited if the paper could tackle noisy, sparse, partially observable point clouds (e.g., from meshes with self-occlusion) that are common in real-world 3D point clouds.

On a related note, I am trying to understand this paper from another perspective: Consider using the node positions from the ground-truth simulator as the point cloud input to this approach, i.e., once in a few frames, the method sees the ground-truth mesh node locations (are these exactly what we want the network to predict?) but without edge information. In this case, I am unsure whether predicting new states is still a challenging problem. Since the point cloud used in this paper seems denser, I guess the problem setup in the paper is even easier. Of course, the authors should feel free to correct me if I misunderstand something.

Regarding the experiment:
- The physics parameters used in the dataset are not very realistic (5000Pa Youngs modulus and {-0.9, 0, 0.49} Poisson’s ratio). A few more physics parameters, e.g., the density of the material and the size of the object, seem missing.
- Fig. 4: Could you show the side view of this scene, e.g., seeing the deformable object from its left?
- Overlaying the point cloud data on the deformable object in Fig. 1 is nice and informative. Is there a similar figure for the other two examples?
- It looks like the point cloud is typically denser than the mesh nodes (600 vs. 81 in B.1 and 850 vs. 361 in B.2), so I guess “voxel-subsample” in the text describing the alpha-shape baseline means downsampling the point cloud to have similar node numbers as the mesh. Why not creating alpha shapes directly from the original point cloud?
- Fig. 6 (b): Does k = 1 mean a point cloud is available in every time step?


**Summary Of The Paper:**

This paper combines graph neural networks with point cloud information to improve the performance of the popular graph network simulators. Standard graph network simulators accumulate prediction errors, causing predicted dynamic states to drift over time. The key idea in this paper is using point cloud information whenever available to counteract such drift and error accumulation. The paper provides three experiments involving 2D and 3D deformable objects to evaluate the proposed approach.

**Summary Of The Review:**

The paper suggested an interesting direction, but I feel the technical method is not ready yet. I also have concerns with the experiments.

---

> ### Author Response · Authors · 2022-11-15
> **Reply to Reviewer ZGJG (Part 1/3)**
>
> Please note that this reply has been split into three parts due to character limits. (Part 1/3)
>
> We thank the reviewer for their constructive comments and detailed feedback, as well as for pointing out technical concerns and unclarities. In the following, we want to address the individual concerns mentioned by the reviewer. We will upload a revised version of the paper in due time and look forward to further comments from the reviewer.
>
> > Regarding the technical method: I think using point cloud to enhance dynamic prediction is a good direction to explore, but I feel the proposed method is not quite ready and the problem setup in this paper is a bit contrived: the point cloud data is generated from the ground truth mesh in a simulator and seems much denser than the mesh nodes (the DoFs in simulation). I would be much more excited if the paper could tackle noisy, sparse, partially observable point clouds (e.g., from meshes with self-occlusion) that are common in real-world 3D point clouds.
>
> We thank the reviewer for their suggestion and we will add additional experiments with partial observability and noisy point clouds in the revised paper, which we will publish by the end of the week. We find that our approach can handle these more realistic and challenging scenarios better than the baseline method. The performance naturally degrades for higher point cloud noise and more occlusion.
>
> Regarding sparsity, we would like to note that more points in the point cloud than mesh nodes is a  realistic assumption, since our point clouds are generated with a normal camera model and then subsampled. We are not aware of a better way of generating the point clouds. Yet, we agree that the experiment with partial and noisy point clouds is very important and will add this to the revision (see above).
>
> We also want to mention that for the existing experiments the point clouds are generated via raycasting, and will clarify this in the revision. The raycasting uses five virtual cameras for the 3d datasets, which follow a realistic setup of cameras around and above the scene and no camera from below. In many cases, this results in a good but not complete coverage of deformable objects with points of the point cloud. E.g., for the Cavity Grasping dataset, the model needs to predict how a cavity is deformed by a robot gripper, but generally does not receive proper point cloud information from the inside of the cavity.
>
> > On a related note, I am trying to understand this paper from another perspective: Consider using the node positions from the ground-truth simulator as the point cloud input to this approach, i.e., once in a few frames, the method sees the ground-truth mesh node locations (are these exactly what we want the network to predict?) but without edge information. In this case, I am unsure whether predicting new states is still a challenging problem. Since the point cloud used in this paper seems denser, I guess the problem setup in the paper is even easier. Of course, the authors should feel free to correct me if I misunderstand something.
>
> We aim to predict the ground-truth mesh node locations, and train on a MSE per node. A scenario where we input the node positions of the ground truth mesh every $k$ steps would simplify the task. However, since there are no point correspondences over time, the network might predict different locations for each mesh node when compared to the real observations due to the different material properties. Then, one observed “point” could be in between two predicted points, making it unclear how to resolve the point correspondence issue.
>
> The use of denser point clouds does not make this setting easier as the dense point cloud does not really contain more information about point correspondences. The points are sampled via raycasting in world space, and thus do not belong to any specific part of the mesh, but can rather be seen as some “interpolation” between mesh vertexes. The main challenge remains the same, i.e. that there are no point correspondences and that the model needs to figure out which point of the point cloud belongs to which vertex in the mesh to do the correction of the mesh nodes.
>
> We agree that these details can further improve the presentation of our approach and will clarify them in the revised paper.

---

> > ### Author Response · Authors · 2022-11-15
> > **Reply to Reviewer ZGJG (Part 2/3)**
> >
> > Please note that this reply has been split into three parts due to character limits. (Part 2/3)
> >
> > > The physics parameters used in the dataset are not very realistic
> > (5000Pa Youngs modulus and {-0.9, 0, 0.49} Poisson’s ratio). A few more
> > physics parameters, e.g., the density of the material and the size of
> > the object, seem missing.
> >
> > We are aware that the chosen material parameters do not represent the full reality. Rather, we want to showcase the capabilities of our method, so we selected these parameters as they displayed the biggest impact on deformation behavior. We agree that other sets of parameters could have achieved similar results, but chose $\[-0.9, 0.49\]$ Poisson’s ratio for simplicity. Please also note that we vary the size of the object for the Cavity Grasping task, where the upper and lower radii of the object are sampled randomly. We will adjust the corresponding parts of the paper to make this clearer and also add the missing physical parameters.
> >
> > > Fig. 4: Could you show the side view of this scene, e.g., seeing the deformable object from its left?
> >
> > We thank the reviewer for their comment and will add visualizations of the scene from additional angles.
> >
> > > Overlaying the point cloud data on the deformable object in Fig. 1
> > is nice and informative. Is there a similar figure for the other two
> > examples?
> >
> > We thank the reviewer for the positive feedback and add a similar illustration for both tasks to the revision.
> >
> > > It looks like the point cloud is typically denser than the mesh
> > nodes (600 vs. 81 in B.1 and 850 vs. 361 in B.2), so I guess
> > “voxel-subsample” in the text describing the alpha-shape baseline means
> > downsampling the point cloud to have similar node numbers as the mesh.
> > Why not creating alpha shapes directly from the original point cloud?
> >
> > We feel that using an alpha shape baseline which creates a mesh with a significantly larger number of nodes, is not really a fair comparison. The reason for that being that the approximation, measured by the IoU, of the ground truth mesh can always be improved by using a mesh with a larger number of nodes. However, this is not really comparable with a fixed sized mesh and leads to a much higher computational effort. The big advantage of our method, however, is the rather compact predicted mesh representation, which can then be used efficiently and which would not be the case with higher resolution meshes. Furthermore, the alpha-shape baseline cannot work with partial point clouds which while our method scales gracefully to this case. Already some experiments reported in the original paper included partial point clouds (points inside of the cavity were missing). Our new experiments with point clouds with less viewpoints confirm that our method works for use cases where the baseline would fail by definition as it does not observe points on the full surface of the object. Moreover, the alpha shape baseline breaks the point correspondence over time when simulating the experiment. This limits the physical understanding of the modelled process, as the evolution of individual particles in the system cannot be observed.
> >
> > > Fig. 6 (b): Does k = 1 mean a point cloud is available in every time step?
> > >
> >
> > Yes. For all figures, $k=1$ means that point clouds are available at each step. We clarify this in the caption of the figure in the revision.
> >
> > > Fig. 1 was a bit confusing when I read the paper for the first time, but most of it
> > became understandable after reading the corresponding sections:
> > It is still unclear to me what solid and dashed lines in Fig. 1 attempt to imply, although it is not a very big deal.
> >
> > We agree that this can be better explained and thank the reviewer for their comment. Solid lines show parts of the system that are active in all time steps, while dashed lines show parts that are only usable with point clouds. Together, they are intended to illustrate the way imputation works. We clarify this in the revision.
> >
> > > Why is there an inference arrow pointing from GNN outputs back to “State at Step t”?
> >
> > The arrow is meant to illustrate the iterative prediction of the subsequent states, which, starting from an initial mesh, is predicted during inference. This is a rollout prediction where the predicted mesh at time t is used as input for the prediction of the mesh at time $t+1$. We change this for clarification in the revision.

---

> > > ### Author Response · Authors · 2022-11-15
> > > **Reply to Reviewer ZGJG (Part 3/3)**
> > >
> > > Please note that this reply has been split into three parts due to character limits. (Part 3/3)
> > >
> > > > One thing that is unclear to me is how the collisions between the rigid and soft bodies are handled in their experiments, and it would be nice if the supplemental materials could include more detailed descriptions about it.
> > >
> > > There are two types of collision handling, that of the ground truth simulator SOFA and the GGNS. SOFA models both bodies as triangular surface meshes that are used for collision detection. The detected collisions are then included in the constraints of the system which are processed together with the other forces to solve the FEM system. Please refer to ([https://hal.inria.fr/hal-00681539/document](https://hal.inria.fr/hal-00681539/document)) for a more thorough explanation.
> > >
> > > In contrast to that, GGNS uses one-hot encoded edges between the rigid and the soft body and feeds the resulting graph into a learned GNN, which in turn computes the dynamics. There is no explicit handling of collisions, the network learns to avoid them and adapts the mesh correspondingly.
> > >
> > > We agree that this can be better explained in the current manuscript and add it to the revised paper.
> > >
> > > > The paper suggested an interesting direction, but I feel the technical method is not ready yet. I also have concerns with the experiments.
> > >
> > > We agree with the reviewer that the presentation of the technical method needed some clarification and hope that the above further explanation will contribute to this. In addition, we would like to refer to the new experiments on noisy and partial point cloud information, which show the applicability of our method in more complex scenarios that are closer to real world data.

---

> > > > ### Comment · Reviewer_ZGJG · 2022-11-26
> > > > **Thank you for the update**
> > > >
> > > > Thank you so much for the detailed explanation and careful revision! I agree with the statement on point correspondence and appreciate the new experiments on noisy data and partial point clouds. Therefore, I will raise my score to 6.

---

### Official Review · Reviewer_vQH9 · 2022-10-24

**Confidence:** 3
**Correctness:** 3
**Technical Novelty And Significance:** 3
**Empirical Novelty And Significance:** 3
**Recommendation:** 8

**Clarity, Quality, Novelty And Reproducibility:**

The authors conducted extensive evaluations across multiple tasks covering Deformable Plate,  Tissue Manipulation, and Cavity Grasping.

The results support the effectiveness of the proposed model.

**Strength And Weaknesses:**

The paper is well presented and motivated. It highlights the imperfection of simulation exhibiting high error accumulation particularly when the initial system state might be incomplete. In response to this, the authors seek to add additional point cloud information to guide the simulation via simple graph combination or fusion and imputation-based training schema. Both quantitive and qualitative results verify the effectiveness of these simple designs.

“Our method, however, omits this explicit representation in favor of a simple one-hot encoding of the type of input edge.”. The authors should have provided an explanation of why they prefer the one-hot encoding.

More insights should have been brought into the paper when the authors make a comparison against the recurrent imputation model. It is suggested to provide some visualization of the simulation results between the recurrent imputation model and GGNS.

It remains unclear to the reviewer the influence of the simulated result on downstream applications in model predictive control and model-based reinforcement learning, which seems to be an obvious limitation to demonstrate the model’s practical value.


**Summary Of The Paper:**

The paper proposes the Grounding Graph Network Simulator (GGNS) that aims to improve the simulation quality by integrating point cloud sensor information for making predictions of the mesh state of deformable objects. An effective imputation training schema is proposed for leveraging the sensor information.  Experimental results demonstrate that the resulting model improves the simulation quality in several deformation prediction tasks, partially when the initial system state is incomplete.

**Summary Of The Review:**

The reviewer tends to give acceptance to the paper given that it addressed the key issue of simulation by introducing a simple solution with additional sensory information.

---

> ### Author Response · Authors · 2022-11-15
> **Reply to Reviewer vQH9**
>
> > The paper proposes the Grounding Graph Network Simulator (GGNS) that aims to improve the simulation quality by integrating point cloud sensor information for making predictions of the mesh state of deformable objects. An effective imputation training schema is proposed for leveraging the sensor information. Experimental results demonstrate that the resulting model improves the simulation quality in several deformation prediction tasks, partially when the initial system state is incomplete.
>
> We thank the reviewer for their constructive comments and detailed feedback, and especially for the precise summary and the positive review. In the following, we want to address the individual concerns mentioned by the reviewer. We will upload a revised version of the paper in due time and look forward to further comments from the reviewer.
>
> > “Our method, however, omits this explicit representation in favor of a simple one-hot encoding of the type of input edge.”. The authors should have provided an explanation of why they prefer the one-hot encoding.
>
> We agree with the reviewer that such an explanation can be improved in the paper and will add it to the revised version. This also includes additional experimental results where we tested both variants of the MGN Baseline with the one-hot encoding and the explicit edge partitioning. Here, we did not find any significant advantage of explicit edge type partitioning over using the one-hot encoding on our tasks. We decided to use the one-hot encoding as it is both conceptually simpler and requires less computational power.
>
> > More insights should have been brought into the paper when the authors make a comparison against the recurrent imputation model. It is suggested to provide some visualization of the simulation results between the recurrent imputation model and GGNS.
>
> We thank the reviewer for the suggestion and agree that such visualizations would strengthen our decision for the choice of the imputation scheme. We will add them to the revision.
>
> > It remains unclear to the reviewer the influence of the simulated result on downstream applications in model predictive control and model-based reinforcement learning, which seems to be an obvious limitation to demonstrate the model’s practical value.
>
> We agree with the reviewer that such downstream applications would further strengthen the practical value of our model. Here, we definitely see potential for future work and we are currently starting to work on it.

---

### Official Review · Reviewer_GZAL · 2022-10-25

**Confidence:** 3
**Correctness:** 3
**Technical Novelty And Significance:** 3
**Empirical Novelty And Significance:** 3
**Recommendation:** 6

**Clarity, Quality, Novelty And Reproducibility:**

This paper seems to provide an original work for grounding graph network simulators, but the work may also be incremental. The paper is well written, and the approach is relatively well explained.

**Strength And Weaknesses:**

+ The studied problem on simulating deformable objects is important for robot manipulation.
+ The idea of applying the imputation-based training scheme seems to be a promising solution to address the problem of incomplete initial system states.

Weakness:

- Technical novelty of the paper is marginal. For example:
-- The complicated dynamics of the deformable mesh seems to be addressed by the previous GNS method, which is inherited by the proposed GGNS.
-- Although it seems to be new for graph network simulators, the implementation of the imputation-based training scheme is straightforward: the proposed approach basically learns two graph networks with and without auxiliary point cloud data, and optimizes them to have the probability during training.


**Summary Of The Paper:**

This paper introduces the grounding graph network simulator (GGNS), which extends the previous graph network simulator by using auxiliary point cloud data to simulate dynamics of deformable objects with incomplete initial system states.

**Summary Of The Review:**

Although novelty of this may need to be further justified, the paper introduces several interesting ideas for grounding graph network simulators that can simulate soft objects.

---

> ### Author Response · Authors · 2022-11-15
> **Reply to Reviewer GZAL**
>
> We thank the reviewer for their constructive comments and detailed feedback, particularly for drawing our attention to the novelty of the method. In the following, we want to address the individual concerns mentioned by the reviewer. We will upload a revised version of the paper in due time and look forward to further comments from the reviewer.
>
> > Technical novelty of the paper is marginal
>
> While GGNS builds and improves upon existing graph network simulators, we want to highlight that it addresses a critical research problem: Current GNS methods are often reliable and faster than classical simulations, but they still have problems in the form of drift and error accumulation over time especially for incomplete initial system states. In order to use such simulations for e.g., robotic tasks, we include sensor information to ground the simulation and compensate for such errors. We want to emphasize that this is particularly relevant for future applications, such as Model-based Reinforcement Learning, which can thus make use of learned general-purpose simulators for the first time. We agree with the reviewer that expressing the novelty of our method can be improved and we address this in the revised version.
>
> > The complicated dynamics of the deformable mesh seems to be addressed by the previous GNS method, which is inherited by the proposed GGNS.
>
> The modelling of the complicated dynamics is indeed addressed by previous GNS method, but previous results show that the performance degrades for multistep rollouts and when the initial system state is incomplete. GGNS tackles these challenges by using point cloud information whenever available to counteract the occurring errors. The effectiveness of our approach is particularly evident in the results of our experiments: quantitatively in Figure 5 and qualitatively in Figure 3&4.
>
> > Although it seems to be new for graph network simulators, the
> implementation of the imputation-based training scheme is
> straightforward: the proposed approach basically learns two graph
> networks with and without auxiliary point cloud data, and optimizes them to have the probability during training.
>
> We thank the reviewer for this comment and agree that the benefits of the imputation-based approach can be better communicated in the current form of the manuscript. We will adapt this in the new revision of the paper and want to highlight that a single GNN is used for the imputation-based approach, both for inputs with and without point cloud data. Although we agree that this approach is   simple, it still provides an innovative idea to include point clouds and observation updates in GNSs and is effective for both quantitative and qualitative results. Also, in comparison to GGNS+LSTM, it is shown that using recurrence does not provide significant benefits, but requires more computational power and training time. We believe that it is more important to present simple ideas instead of more complex architectures such as recurrent structures that would be easier to sell in terms of novelty but do not provide significant performance benefits.
>
> > Although novelty of this may need to be further justified, the paper introduces several interesting ideas for grounding graph network simulators that can simulate soft objects.
>
> We thank the reviewer for their feedback and agree that the novelty of our work can be stated more clearly and we adapt this in the revised version. For the relevant details, we refer to the detailed questions and answers on the presentation of the novelty above.

---

### Comment · Area_Chair_sdrw · 2022-11-15
**Please engage before the author-reviewer discussion closes**

Dear authors and reviewers,

The first phase of the discussion period is about to close on November 18.

For authors, please make sure to submit your rebuttal by the deadline. Leave some time for the reviewers to read it and respond while you are still allowed to further engage with them. Interactions between authors and reviewers are very important for the quality of the review process, so please make sure to engage.

For reviewers, please try to acknowledge and respond to the authors' rebuttal while the discussion period is still open for them to further interact with you.

Thank you for your participation in the review process!

Best,
The AC

---

### Author Response · Authors · 2022-11-18
**Paper Revision and Changelog**

Dear reviewers,

attached you find a revised version of our paper. For easy reference, we color-coded all changes by the reviewer who suggested them:

- General changes: green
- Reviewer **GZAL**: purple
- Reviewer **vQH9**: orange
- Reviewer **ZGJG**: blue

We briefly list the changes made in the revised paper, denoting the reviewer(s) whose comments motivated these changes in parenthesis.

- Renamed Chapter 4 to Grounding GNS for improved consistency
- Added subsection B.1 “Point Cloud Generation” to Appendix B for improved structure
- Improved the description of the novelty and relevance of the presented work in introduction (**GZAL**)
- Added more details on the imputation-based training scheme and its advantages over the recurrent model in the method section (**GZAL, vQH9**)
- Clarified choice of one-hot encoding vs. explicit edge partitioning, moved corresponding section to experiments (**vQH9**)
- Added experiments on one-hot encoding vs. explicit edge partitioning in Appendix D (**vQH9**)
- Added visualization of the simulation results of GGNS+LSTM in Appendix C (**GZAL, vQH9**)
- Added experiments on noisy point clouds for Deformable Plate dataset in Appendix D (**ZGJG**)
- Added experiments on partial observable point clouds for Cavity Grasping dataset in Appendix D  (**ZGJG**)
- Clarified the point cloud generation via raycasting in experiments and further details in  Appendix B.1 (**ZGJG)**
- Clarified the challenge of missing correspondences between points in point clouds in introduction and Appendix B.1 (**ZGJG**)
- Clarified choice of material properties in Appendix B (**ZGJG**)
- Added different viewing angles for Figure 4 in Appendix C (**ZGJG**)
- Added additional figures overlaying the point cloud data on the deformable object for tissue and plate tasks in Appendix C (**ZGJG**)
- Added additional insights on the Alpha shape baseline in results (**ZGJG**)
- Clarified meaning of $k=1$ in Figure 6(b) (**ZGJG**)
- Improved clarity and updated description of Figure 2 (**ZGJG**)
- Added Collision Handling description in Appendix B.3 (**ZGJG**)

We want to thank the reviewers again for their helpful comments. We look forward to hearing about any concerns that are left unaddressed by the revised paper, as well as to additional feedback.

---

### Decision · Program_Chairs · 2023-01-20

**Decision:**

Accept: poster

**Justification For Why Not Higher Score:**

This work is solid and well-executed. However, I wouldn't recommend a higher score as it is not groundbreaking either -- it is an extension of the Graph Network Simulator framework.

**Justification For Why Not Lower Score:**

Technically solid. All reviewers' concerns have been addressed.

**Metareview: Summary, Strengths And Weaknesses:**

All reviewers unanimously recommend acceptance (6-8-6). The author-reviewer discussion was productive and the authors implemented many improvements to address the reviewers' concerns. No major concerns remain. The paper is ready for publication.

**Note From Pc:**

if the above contains the word "oral" or "spotlight" please see: "oral" presentation means -> notable-top-5% and "spotlight" means -> notable-top-25%. As stated in our emails, we are disassociating presentation type from AC recommendations

**Summary Of Ac-Reviewer Meeting:**

N/A